# Low-Cost GNSS-R Altimetry on a UAV for Water-Level Measurements at Arbitrary Times and Locations

**DOI:** 10.3390/s19050998

**Published:** 2019-02-26

**Authors:** Kaoru Ichikawa, Takuji Ebinuma, Masanori Konda, Kei Yufu

**Affiliations:** 1Research Institute for Applied Mechanics, Kyushu University, Kasugai 816-8580, Japan; yufu@riam.kyushu-u.ac.jp; 2College of Engineering, Chubu University, Kasugai 487-8501, Japan; ebinuma@isc.chubu.ac.jp; 3Graduate School of Science, Kyoto University, Kyoto 606-8502, Japan; konda@kugi.kyoto-u.ac.jp

**Keywords:** GNSS reflectometry, altimetry, unmanned aerial vehicle, lake water level

## Abstract

Together with direct Global Navigation Satellite System (GNSS) signals, the signals reflected at the water surface can be received by an unmanned aerial vehicle (UAV). From the range difference between two GNSS signal paths, the height of the UAV above the water level can be geometrically estimated using the weighted least squares method, called GNSS reflectometry (GNSS-R) altimetry. Experimental low-cost GNSS-R altimetry flights with a UAV were conducted at the coast of Lake Biwa, Japan. Although the height estimated by the GNSS-R altimeter included large short-term noises up to 8 m amplitude, it agreed well with the UAV altitude measured by the post-processed kinematic positioning. By selecting better weight functions in the least square method and using sufficient temporal averaging, the GNSS-R altimetry achieved accuracy in the order of 0.01 m if a sufficient number of GNSS satellites with high elevation angles were available. The dependency of the results on the weight functions is also discussed.

## 1. Introduction

Sea surface height is a fundamental variable in physical oceanography that is used in various studies, such as studies of waves, tides, tsunamis, geostrophic currents, and climate change. Satellite altimeters have been widely used to monitor sea surface height in open oceans [1]. However, in coastal areas, conventional satellite altimeters do not perform as designed, because the microwave reflectance within a footprint tends to be inhomogeneous in these areas due to the presence of land, slicks, and calm seas in semi-closed bays [2,3,4]. In addition, the spatial and temporal resolutions of along-track satellite altimeters are not high enough to describe coastal phenomena and/or variations of land water, such as lakes, rivers and deltas. Although new experimental wide-swath altimeters, i.e., SWOT (Surface Water and Ocean Topography) [5] and COMPIRA (Coastal and Ocean measurement Mission with Precise and Innovative Radar Altimeter) [6], are planned to provide better spatial resolution and coverage, additional frequent height observations are necessary to satisfy the requirements of the temporal resolution.

The Global Navigation Satellite System (GNSS) is a most plausible practical candidate to measure the height in coastal areas or on land. Precise GNSS positioning would provide a reasonable accuracy of height observations at a point [7]. For the use of oceanography, however, a wide spatial coverage is also required, and therefore GNSS receivers can be mounted on drifters [8], ships [9], unmanned aerial vehicles (UAVs), aircraft [10] and satellites [11]. For those receivers on a platform, the vertical distance of the receiver antenna from the sea surface (h) should be separately determined together with the GNSS antenna height, $H_{a}$, to obtain the sea surface height, $H_{S}$ (Figure 1). 

The height of the antenna h can be determined by measuring the excess path length of the reflected GNSS signals with respect to the direct signals (Figure 1), i.e., GNSS reflectometry (GNSS-R) altimetry [12,13]. The excess path length is either directly determined by measuring the temporal delay of the reflected signals [10,11,14,15] or indirectly estimated from interferometric cyclic modulations of the received power (i.e., signal-to-noise ratio; SNR) induced by phase differences between the direct and reflected GNSS signals [16,17]. The latter method requires only a classical GNSS receiver with a single right-handed circular polarization (RHCP) antenna. However, since the reflected GNSS signals tend to be left-handed circular polarization (LHCP) at higher elevation angles [16], the use of this method is limited and can be used only when GNSS satellites at lower elevation angles are available and when other factors altering SNR, such as wind speeds, are temporally unchanged. On the other hand, the former method is applicable at any time and location, although it uses both RHCP and LHCP antennas to separately receive the direct and reflected GNSS signals. Ideally, these two GNSS signals should be recorded by synchronized receivers to avoid each receiver being contaminated by clock errors, although synchronized receivers are generally more expensive than classical receivers. In the present study, low-cost GNSS-R altimetry, applicable at arbitrary times and locations, was examined by mounting two classical receivers on a UAV.

We conducted experimental GNSS-R altimetry flights on a UAV at Lake Biwa, Japan. The basis of the GNSS-R altimetry method and details of the experiments is described in the next section. The results are presented in Section 3, followed by discussion of the methods in Section 4, with a short summary in Section 5. 

## 2. Materials and Methods

### 2.1. GNSS-R Altimetry Method

With knowledge of the relative position between a UAV and a given GNSS satellite, and the assumption of a flat, non-tilted reflection surface, the difference of the geometric ranges between the direct and reflected signal paths, d, can be expressed as follows:(1)d=2hsin(E),
where E is the elevation angle of the GNSS satellite, and h is the altitude of the UAV from the water surface (Figure 1).

On the other hand, since a measured pseudo range of the direct signal path, $L_{D}$, is subject to unknown clock errors of the satellite ($\Delta t_{S}$), and the receiver ($\Delta t_{D}$), the actual range, $r_{D}$, will satisfy the following equation:(2)$$L_{D}=r_{D}+c\left(\Delta t_{D}-\Delta t_{S}\right)+\Delta \epsilon_{D},$$
where c is the speed of light, and $\Delta \epsilon_{D}$ is an observational range noise. Similarly, the actual and measured ranges of the reflected path, $r_{R}$ and $L_{R}$, respectively, will satisfy the following equation:(3)$$L_{R}=r_{R}+c\left(\Delta t_{R}-\Delta t_{S}\right)+\Delta \epsilon_{R},$$
where $\Delta t_{R}$ is an unknown clock error of the receiver for the reflected signal path and $\Delta \epsilon_{R}$ is an observational range noise of the reflected signal. Equations (2) and (3) provide:(4)$$\begin{array}{c} L_{R}-L_{D}=\left(r_{R}-r_{D}\right)+c\left(\Delta t_{R}-\Delta t_{D}\right)+\left(\Delta \epsilon_{R}-\Delta \epsilon_{D}\right),\\ \Delta L\;=d+\;c\Delta T+\Delta \epsilon . \end{array}$$
where ΔL is the measured pseudo range differences, ΔT is an unknown floating clock error difference between two receivers, and Δϵ is an unknown observational range noise.

In typical open sky conditions, more than a few GNSS satellites are observable at the same time, namely with the common ΔT. At a given time, the single differential range observation from the i-th GNSS satellite can be described as follows:(5)$$\Delta L_{i}=2h\sin\left(E_{i}\right)+c\Delta T+\Delta \epsilon_{i}=\left[\begin{array}{cc} 2\sin\left(E_{i}\right) & 1 \end{array}\right]\left[\begin{array}{c} h\\ c\Delta T \end{array}\right]+\Delta \epsilon_{i}.\;$$From i=1, 2, …, N satellite observations, the following simultaneous equations can be formulated, taking account of the reliability of each observation by a weight $w_{i}$:(6)$$\left[\begin{array}{c} w_{1}\Delta L_{1}\\ \vdots \\ w_{N}\Delta L_{N} \end{array}\right]=\left[\begin{array}{cc} 2w_{1}\sin\left(E_{1}\right) & w_{1}\\ \vdots & \vdots \\ 2w_{N}\sin\left(E_{N}\right) & w_{N} \end{array}\right]\left[\begin{array}{c} h\\ c\Delta T \end{array}\right]+\left[\begin{array}{c} w_{1}\Delta \epsilon_{1}\\ \vdots \\ w_{N}\Delta \epsilon_{N} \end{array}\right],\;$$
which can be expressed as the following matrix and vector formula:(7)y=Ax+ϵ.Using the least squares method, the following equation provides the optimal solution of the unknown parameter vector, $\hat{x}$, at the given time:(8)$$\hat{x}={\left(A^{T}A\right)}^{-1}A^{T}y.$$

The weight in Equation (6) is introduced in order to account for differences of reliability for each observation with different GNSS satellites. In general, a GNSS satellite with a lower elevation angle, $E_{i}$, is less reliable, since the reflected GNSS signal has less of the LHCP component that was measured by the UAV in this study. In addition, the range difference, $d_{i}$, in Figure 1 became larger as $\sin\left(E_{i}\right)$ increased, thus the noise ratio of each observation in Equation (4), or $d_{i}/\left(c\Delta T+\Delta \epsilon_{i}\right)$, depends on $\sin\left(E_{i}\right)$. Therefore, we chose the weight function $w^{s}=\sin\left(E\right)$, together with the no weight case $w^{no}=1$.

In Figure 1, the GNSS-R altimetry assumed reflections from a flat, non-tilted water surface, which would not be realistic and would produce observation noise Δε. Under the presence of tilting slopes, the actual reflection point would shift from the ideal geometric location. With the slight tilt of the flat water surface in the radial direction, θ, the slant range from the actual reflection point to the antenna,$\;h/\sin\left(E_{i}+2\theta \right)$, differs from the assumed ideal range, $h/\sin\left(E_{i}\right)$, by $2h\theta /\sin\left(E_{i}\right)\tan\left(E_{i}\right).$ This means that the effect of tilting slopes would decrease as $\sin\left(E_{i}\right)\tan\left(E_{i}\right)$ increases, so that we introduce another weight function as $w^{st}=\sin\left(E\right)\tan\left(E\right)$ in order to further account for noises caused by tilting slopes of the water surface.

### 2.2. Experimental Equipment

In our experiments, upward and downward lightweight GNSS antennas were mounted on a quadcopter (DJI Phantom 2 Vision+) to receive the direct and reflected GNSS signals, respectively. In order to effectively receive the reflected signals, an LHCP antenna (Antcom 4G15L-A-XS-1) was used as the downward antenna (Figure 2b), while an RHCP antenna (Tallysman TW4721) was selected for the upward antenna (Figure 2a). Both antennas have reasonable LHCP-versus-RHCP gain ratios larger than 15 db down to 10° elevation angle, so we limited the elevation angle of GNSS satellites to be larger than 10°. The vertical distance between the two antennas was 0.15 m. The GNSS signals were recorded on board at the 5 Hz rate by two independent classical receivers (ublox NEO-M8T).

Unfortunately, a lightweight GNSS LHCP antenna was not available at the time of the experiments, so the LHCP antenna used was only for the Global Positioning System (GPS) L1 band (1575.42 ± 12 MHz). Therefore, we needed to limit our analysis to the GPS consternations in the present manuscript, although the inclusion of weakly-received BeiDou signals will be discussed in Section 4.

### 2.3. Flight Experiments

Two experimental flights were conducted at the western coast of Lake Biwa (35.319°N, 136.077°E), Japan (Figure 3a) on 7 January 2017, during the period 12:00–14:00 JST. The UAV moved straight upward at the coast, and hovered at approximately 120 m height for about four minutes during the first flight (see Figure 4a). Meanwhile, during the second flight, the UAV again hovered, for approximately 1 minute at different altitudes (i.e., 65 m, 90 m, 120 m, 80 m, and 50 m; see Figure 4b). In both flights, wind speeds were less than 1.5 ms^−1^. 

We deployed a geodetic 1-Hz GNSS receiver (Hitz NetServer RE) as an in situ base station (Figure 3b). Using the RTKLIB [18] post-processed kinematic (PPK) positioning method, the height of the in situ base station was determined to be 123.21 m (with 0.074 m standard deviation) at the WGS84 coordinates, with reference to Adogawa station, which represented a baseline of ~1 km; the Adogawa station is one of the stations in the Japanese GNSS Earth Observation Network System (GEONET) that is operated by the Geospatial Information Authority of Japan. The height of the UAV, namely $H_{a}$ in Figure 1, was determined by the PPK positioning method by referring to our base station [19,20]. 

Since both flights were conducted at the western boundary of the lake, in the following GNSS-R analysis, we eliminated signals from GPS satellites located to the west of the UAV, whose reflection points were expected on the land surface rather than on the water surface. Due to this limitation, the number of GPS satellites in use, or N in Equation (6), did not exceed five.

## 3. Results

The temporal deviation from the mean water level was −0.10 m during the observation period at the closest Oomizo water level observatory (35.2901°N, 136.0161°E; operated by the Ministry of Land, Infrastructure and Transport, Japan), whose WGS84 mean water level height was 120.94 m. Using this water-level height, $H_{S}$, and the measured altitude of the UAV, $H_{a}$, the measured altitude of the UAV from the water level ($H=H_{a}-H_{S}$) was calculated to compare with the height estimated by the GNSS-R method, namely h in Equation (6).

Figure 4 shows a time series of the measured and estimated altitude of the UAV, H and h, respectively, with different weight functions. In general, the measured ascending, hovering, and descending statuses of the UAV were well represented in the estimated altitudes. On the other hand, the estimated altitude h included large, short-term variations; they were especially large in the first flight when the weight function $w^{st}$ was used.

Figure 4 also indicates that deviations were larger during hovering periods than during ascending and descending periods. In general, the attitude of the UAV was more stable during ascends and descends than during hovering, and the sway motions of the antennas would affect observations of the range difference [15]. However, further studies in the future are required, since the attitude of the UAV was not measured in the experiments.

Statistics of the difference of the measured and estimated heights are summarized in Table 1. In these comparisons, the vertical distance between the upward and downward antennas was adjusted as H−h−0.15. In the calculation of the statistics, data were not used when the UAV altitude was below 10 m to eliminate possible contamination by the land reflections during the takeoff and landing processes. As seen in Figure 4a, large short-term variations in the first flight with $w^{st}$ resulted in a large root-mean-squared (rms) difference of 8.09 m. Similarly, in the second flight, the weight $w^{st}$ provided the largest rms difference among the three weights. On the other hand, the mean difference, or the accuracy of the GNSS-R altimetry results, was the best with the weight $w^{st}$.

When comparing the first and second flights, it can be seen that the GNSS-R altimetry performed better in the second flight, for all weight functions. As seen in the mean sum of $w^{no}$, the number of available GPS satellites was almost always four in both the first and second flights. However, the elevation angles of the GPS satellites were generally higher in the second flight, as indicated by the sum of $w^{s}$, which indicated that the GNSS-R altimetry performed better when more GNSS satellites with higher elevation angles were available.

When the comparisons were limited to higher and lower altitudes, discrepancies of the weight functions were enhanced. For the uniform weight $w^{no}$, the GNSS-R altimeter performed better at the lower altitudes in both the first and second flights, and the mean difference became −1.27 m or −0.03 m, respectively, if the UAV altitude was below 60 m. For the weight $w^{s}$, the lower UAV altitude had similar results, which were better for both flights. Although, in the second flight, the absolute values of the mean differences, i.e. 0.38 m and 0.21 m, were not significantly different. On the contrary, for the weight $w^{st}$, the absolute values were not significantly different for the first flight. For the second flight, the performance of the GNSS-R altimetry was better for the higher UAV altitude, as there was only a −0.02 m difference. If the best weight functions are selected for both higher and lower UAV altitudes in the better-quality second flight, the mean discrepancy between the measured and estimated heights archives an accuracy of the order of 0.01 m, namely -0.02 m and -0.03 m, which is well compatible with that of the 5 Hz PPK positioning of the UAV height ($H_{a}$) [7].

Note also that, in all cases with any weight functions in both flights, the mean differences became more negative when the UAV’s altitude was limited in height. This may suggest that the present GNSS-R altimetry tends to overestimate (or underestimate) at higher (or lower) altitudes. 

## 4. Discussion

In practice, the weighted least squares method is subject to a tradeoff between the number of observations and the quality of the data. Generally, noise in each observation tended to remain if the number of observations was not sufficient for the stable statistical averaging. On the contrary, in the present analysis, observations by GPS satellites with lower elevation angles did not contribute effectively in Equation (8), except for the uniform weight $w^{no}$. The decrease of the number of effective observations would be most (or least) prominent for the case of $w^{st}$ (or $w^{no})$, and thus the estimated height became more (or less) noisy by unsuppressed Δϵ, as in Table 1a. Meanwhile, these noises could be further smoothed out by temporal averaging, so that the best mean estimated height would be achieved in the case of $w^{st}$.

As described in Section 2, the weight $w^{st}$ was introduced in order to account for changes in the slant ranges caused by displacement of the ideal and actual reflection points. Since these changes also depend on h, this weight would be more important at higher UAV altitudes. In fact, statistics of $w^{st}$ are the best (or worst) among three weights at the higher (or lower) UAV altitudes. These confirm that the effect of displacement of the reflection points caused by tilting slopes are significant for the higher UAV altitudes, and also suggest that $w^{st}$ would have improperly overweighted observations by GPS satellites with higher elevation angles when the altitude of the UAV is low. 

The displacement of the reflection points due to tilting slopes would monotonically increase the measured path lengths ($L_{R})$ from the assumed path length ($r_{R}$), since the reflection path length is shortest at the geometric reflection point. Therefore, the obtained h would be overestimated by using increased $L_{R}$, especially at the higher UAV altitudes. This would explain the negatively-larger tendency of the mean difference at the higher UAV altitudes in all cases seen in Table 1b,c. Since this systematic overestimated tendency cannot be averaged out by temporal averaging, the low-cost GNSS-R altimetry would perform better in the lower UAV altitudes to avoid the effect of tilting slopes, which would be especially severe for satellites with lower elevation angles.

At the lower UAV altitudes (Table 1c), $w^{s}$ provides stable statistics in both flights, whereas significant discrepancy between two flights are found for $w^{no}$. This would suggest that $w^{s}$ can provide independent results from selections of available satellites, although the statistics of $w^{no}$ would be better if enough numbers of high-elevation satellites are available. 

In the present analysis, all GPS satellites located to the west were eliminated to avoid contaminations by reflections at the land surface. Therefore, more satellites would be available if the location of the UAV is properly selected away from the coast. In addition, the use of GNSS satellites rather than GPS would also increase the number of satellites with better elevation angles. Although the received signal was quite weak (35dBHz) due to the antenna mismatch in the present experiments, the signals from the BeiDou constellations were recorded during the flights. Unfortunately, most of the BeiDou satellites were also located to the west, so that only one satellite (C04) was available for the present analysis. The statistics are summarized in Table 2 for the case that includes the BeiDou C04 signals. The results with $w^{no}$ were generally improved from Table 1, but, in contrast, for the other weight functions the results were worse than those in Table 1. Since the elevation angle of the BeiDou C04 (approximately 46°) was higher than any GPS satellites, the noisy BeiDou data was improperly overloaded for the weight functions $w^{s}$ and $w^{st}$. This would suggest that the weight function should be carefully selected, especially when different GNSS constellations are mixed, and also that both the number and quality of the data are necessary in order to improve the results.

In Equation (6), the floating clock error difference between two receivers (ΔT) was also estimated together with the height h. Although the temporal changes of the estimated ΔT were generally gradual (Figure 5), there was a large gap at 81.8 min in the second flight. In addition, the values of ΔT for the two flights were very different. In the present analysis, Equation (6) was solved for each 5 Hz observation, since ΔT would be different at another observation time. However, if we recursively modified the observed range difference ΔL by the estimated ΔT, the observations at adjacent times would be identical except for the noise errors Δϵ, as if synchronized receivers were used, so that the number of data used in the least squares method would be increased, by using a longer duration of 5 Hz observations. Since h is regarded constant in the least square method, the duration would depend on both the UAV movements and the sea state, which would be further studied in future.

Sea surface height (SSH) in coastal areas within 5–10 km of coastlines has not been observed by satellite altimeters [4], nor by coastal tide gauges, although it was observed by static GNSS-R analysis [21]. Nadir-looking satellite altimeters provide no SSH observations in gaps between subsatellite tracks, which are approximately 300 km for Jason series. Furthermore, repeat cycles of satellites (e.g., 10 days for Jason series) are too long for coastal studies, as more frequent observations are necessary. Additional observations to fill these gaps in space and time will be obtained by the present method, since coastal SSH can be observed at arbitrary times and locations. In addition, the present method can provide observations of the water level of lakes or ponds that are difficult to access, which will be used to estimate changes of their water volume. Note also that a large number of UAVs can be constellated because of the low-cost nature of the present method, which would enable the provision of water level maps with high resolution and wide coverage that could be used to validate wide-swath altimeters, such as SWOT. However, in the presence of larger wave heights, the results of the present analysis could be altered. For example, the weight $w^{st}$ accounts tilting slopes of the water surface, but other impacts of waves, such as changes of GNSS path length induced by wave crests and troughs, could be important for applications in coastal areas or open oceans. Further studies are necessary for practical oceanic applications.

## 5. Summary

Low-cost GNSS-R altimetry on a UAV was examined by mounting LHCP and RHCP antennas and two classical GNSS receivers on a UAV. Although the single 5 Hz observations included large errors, estimates with centimeter-order accuracy were achieved for the temporal mean. The results depend on the weight functions in the least square method and the available number of GNSS satellites of better quality, so that further studies are necessary for the use in other areas and conditions, e.g., rough seas. Nevertheless, the ability to measure the water level at arbitrary times and locations is promising for various researches such as coastal oceanography and limnology, and also for disasters monitoring including surges, tsunamis and floods.

## Figures and Tables

**Figure 1 sensors-19-00998-f001:**
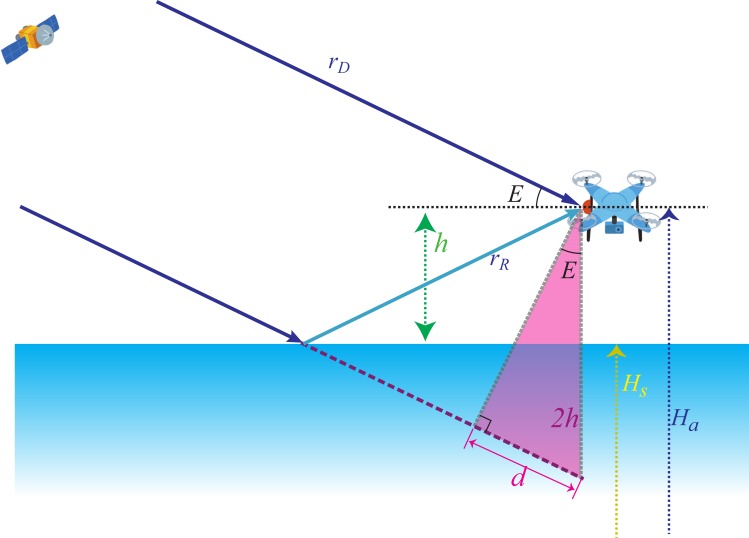
Schematic figure of GNSS-R altimetry on a UAV.

**Figure 2 sensors-19-00998-f002:**
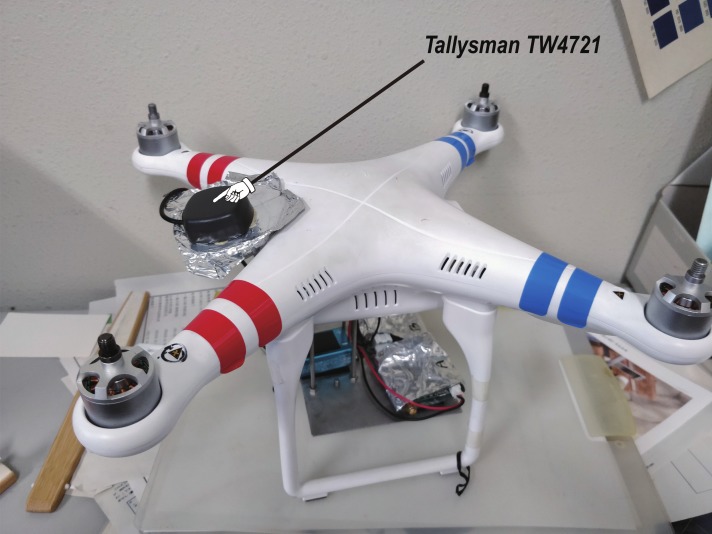
The GNSS-R altimeter used in the experiments: (**a**) Top view; (**b**) Bottom view.

**Figure 3 sensors-19-00998-f003:**
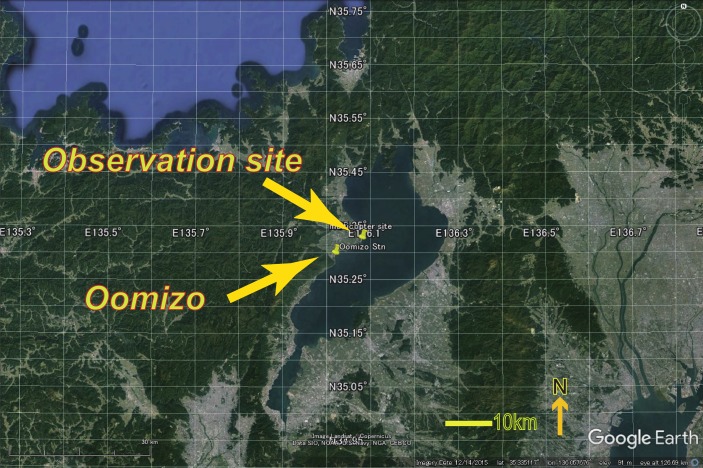
(**a**) Locations of the observation site and Oomizo water level observatory at the western coast of Lake Biwa; (**b**) snapshot of the observation site. The antenna height of the in situ base station was 2.0 m.

**Figure 4 sensors-19-00998-f004:**
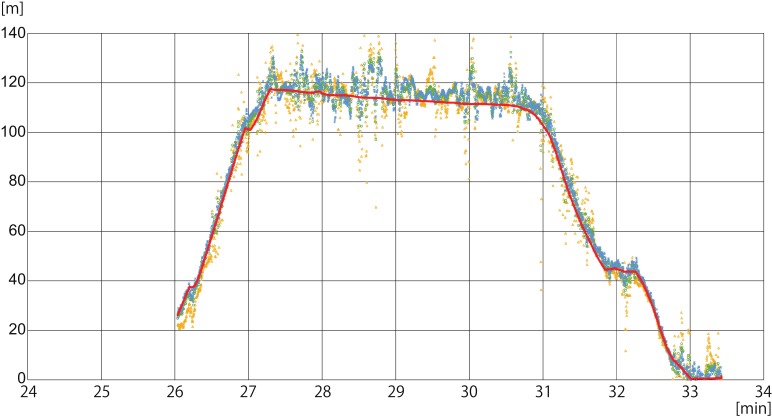
(**a**) Time series of the measured altitude of the UAV, H (red curve), and the estimated heights h with various weight functions in the first flight experiment; blue crosses for $w^{no}$, green circles for $w^{s}$, and yellow triangles for $w^{st}$. The abscissa is the elapsed time in minutes from noon on 7 January 2017 (local JST); (**b**) the same in the second flight experiment.

**Figure 5 sensors-19-00998-f005:**
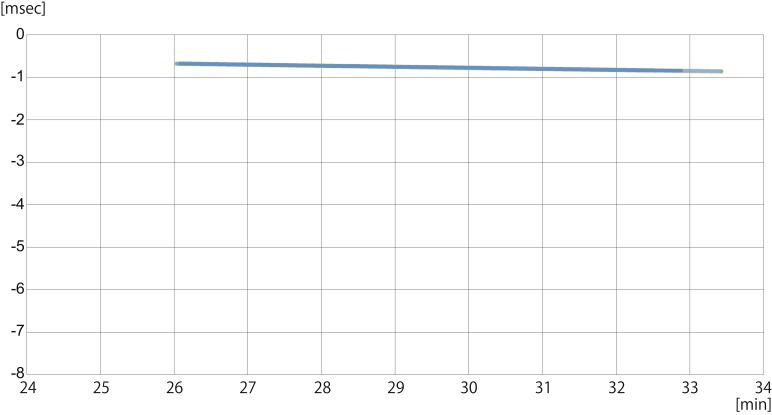
(**a**) The same as Figure 4a, but for the floating clock error difference between two receivers, ∆T; (**b**) the same in the second flight experiment.

**Table 1 sensors-19-00998-t001:** Statistics of the differences between the measured and estimated heights; number of 5 Hz observations, mean and rms height differences, and mean of the sum of the weight values. The statistics were calculated for three cases; (**a**) all data above 10 m altitude; (**b**) selected data higher than 100 m altitude; and (**c**) selected data between 10 m and 60 m altitude.

		First Flight			Second Flight		
Weight Function		$w^{no}$	$w^{s}$	$w^{st}$	$w^{no}$	$w^{s}$	$w^{st}$
(**a**) All (above 10 m)	Number	2005			2499		
	Mean diff. [m]	−2.78	−2.37	−0.98	−0.72	−0.60	−0.37
	Rms diff. [m]	3.81	3.98	8.09	3.32	4.98	7.11
	Mean $\sum_{i=1}^{N}w_{i}$	3.95	1.87	2.00	3.99	2.14	4.56
(**b**) High altitude (above 100 m)	Number	1234			706		
	Mean diff. [m]	−3.35	−3.13	−2.23	−0.68	−0.38	−0.02
	Rms diff. [m]	4.43	4.40	8.80	3.73	5.37	7.22
	Mean $\sum_{i=1}^{N}w_{i}$	3.93	1.87	2.01	3.98	2.14	4.61
(**c**) Low altitude (10 m–60 m)	Number	497			565		
	Mean diff. [m]	−1.27	−0.46	2.14	−0.03	0.21	0.51
	Rms diff. [m]	2.05	2.50	5.52	3.27	4.83	6.66
	Mean $\sum_{i=1}^{N}w_{i}$	4.00	1.86	1.96	4.00	2.11	4.22

**Table 2 sensors-19-00998-t002:** As for Table 1, but including the BeiDou C04 satellite.

		First Flight			Second Flight		
Weight function		$w^{no}$	$w^{s}$	$w^{st}$	$w^{no}$	$w^{s}$	$w^{st}$
(**a**) All (above 10 m)	Number	2005			2499		
	Mean diff. [m]	−1.62	−3.82	−8.94	−0.02	−1.57	−3.91
	Rms diff. [m]	4.43	5.11	17.33	3.07	5.24	7.65
	Mean $\sum_{i=1}^{N}w_{i}$	4.87	1.87	2.00	4.95	2.81	5.23
(**b**) High altitude (above 100 m)	Number	1234			706		
	Mean diff. [m]	−2.16	−4.68	−10.35	−0.04	−1.29	−3.34
	Rms diff. [m]	5.35	5.89	21.23	3.63	5.367	8.64
	Mean $\sum_{i=1}^{N}w_{i}$	4.80	2.48	2.60	4.81	2.72	5.17
(**c**) Low altitude (10 m−60 m)	Number	497			565		
	Mean diff. [m]	−0.46	−1.39	−3.78	0.37	−0.28	−1.54
	Rms diff. [m]	2.06	2.52	5.44	2.90	5.26	7.46
	Mean $\sum_{i=1}^{N}w_{i}$	5.00	2.56	2.65	5.00	2.81	4.90

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
