# Peer review of "Low-Cost GNSS-R Altimetry on a UAV for Water-Level Measurements at Arbitrary Times and Locations"

_sensors, 2019, doi:10.3390/s19050998_

Reviewer 1 Report

Dear authors,

This article examines an interesting research topic for Remote/proximal Sensing application on water surface height measurements, in coastal areas, by using an interesting methodological approach based on the GNSS receivers mounted on and unmanned aerial vehicles (UAVs), and comparing the obtained result (using different weight functions) with on site observation.

This methodology can open an interesting research activity on monitoring the spatial/temporal variation of water stored in land reservoir (by integrating the information on water height with those on the water surface - coming for example from multispectral observation from Sentinel-2/Landsat 8 satellite - and bathymetric or surface/volume diagram data), which represent a fundamental information for water use planning in the context of climate changes scenarios.

Author Response

Thank you very much for your positive comments that encourage us very much.

We have further modified our documents to increase in several points, including

+ Title has been changed to enhance advantages of our method 

   (low cost, and no-limitation of locations and times)

+ Include the results for Delta T

+ Include additional analysis with BeiDou signals, 

   although they were not well received 

Reviewer 2 Report

As a general comment, I would like to congratulate the authors for the work. It is a very detailed and a high-quality work as it is the purpose of every scientific work. But, for me the results are poorly analyzed, the Section 4 integrates weight function definition that may need to be included in Section 2. The discussion must be improve (see my comments): discuss the efficiency of this method versus others GNSS-R altimetry technics and also radar altimeter (Jason 2 and 3, SARAL, Sentinel 3, etc.). 

Another critical point is: Why the authors have limited the analysis to GPS constellation, the use of other constellations increase the number of visible satellites and thus the height retrieval quality. 

Last point, the author describe a weight function call Wst define for waves but they test it on a lake without waves (see figure 3b)... This could explain why Wst RMS difference is upper than the other functions (Table 1) so it is difficult to highlight the efficiency of this function.

Overall, the paper is well written and organized, though I have tried to point out most of the grammatical errors below, (see comments).

However, there are some aspects that I would like to discuss in detail about the work, as I consider that with a different approach it might be of much more worth to the scientific community. 

For all of these raisons I recommend Major Revision

This paper has three main new aspects:

             the first one shows the theoretical bases for GNSS-R altimetry using two classical GNSS receivers (one for the uplooking RHCP antenna one for the downlooking LHCP antenna), the single differential range observation is used to retrieve the height above the water surface;

             the second one shows the UAV experimental platform used : UAV description, GNSS antenna positions, polarization (Right Hand circular for direct path,   Left Hand for the reflected one);

            the last one presents the result for different weight matrix types of measurements (uniform  : Wno, Ws depending on sin(elevation), Wst depending on sin(elevation) x tan(elevation))

Main remarks to discuss: 

For the comments (see below): m is minor and M moderate/major comments

M: The title does not well described the content of this paper and must be improved; why do you want to do GNSS-R altimetry? What is your goal? This should be made clearly in the title and in the abstract.

M: In your introduction: Your second part (line 32 to 40) is to short and does not give a clear state of the art of GNSS and GNSS-R altimetry. For example, you have forgotten several reference papers:

              Frederic Frappart for tidal Bore measurement using classical GNSS receiver

High rate GNSS measurements for detecting non-hydrostatic surface wave. Application to tidal bore in the Garonne River, in European Journal of Remote Sensing, (2016)

            Manuel Martin-Neira:

            A PASSIVE REFLECTOMETRY AND INTERFEROMETRY SYSTEM (PARIS): APPLICATION TO OCEAN ATIMETRY, in ESA J., (1993)

            Laurent Lestarquit: 

            REFLECTOMETRY with an open-source software GNSS receiver: use case with carrier phase altimetry, in IEEE JSTARS (2016)

            Kristin Larson:

           Coastal sea level measurements using a single geodetic GPS. in Advances in Space Research (2012).

 m: line 27 you can after lakes, rivers,  add " and deltas "

 m: line 32 you can suppress " including …(GPS)"           

The Material and Methods section is well described, I have some remarks/questions 

For 2.1 GNSS-R altimetry method

M: Instead, I will begin by presenting the three main GNSS-R altimetry methods: 

             the first one presented by Lestarquit et al. (2016) that use two antennas and a specific waveform receiver;                

             the second one described by Larson (2012) that use only one antenna with a classical GNSS receiver;

             the third one described herein that use two antennas and two classical receivers as it was done by Roussel et al. in 2016 (see below) but for moisture retrieval.

             Ref: Multi-scale volumetric soil moisture detection from GNSS SNR data, Roussel N. et al., 3rd IEEE International Workshop on Metrology for Aerospace (MetroAeroSpace) , Florence, ITALY, 2016, in IEEE METROLOGY FOR AEROSPACE (METROAEROSPACE), p. 573-578 

              In this part it is necessary you discuss a little bit the weight function that you used (in section 4 you describe it but you have to describe it here)  if I understand Wno  corresponds to no weight function, you consider in that case all the measurements have the same quality (as first approximation ok!). 

               For Ws function: in that case, you write:  weight do not depend on the elevation angle but from 1/ sin(elevation) and you multiplied a weight to reduce these effects. For me, a more clear explanation could be associated to the wave path through the atmosphere, for high elevation satellite this path is short and we have a less noise so the measurement has a better quality and the weight is close to 1. For low elevation the noise level is upper due to a larger path through the atmosphere, and then the weight function as a weaker value.

                But for the last function could you explain a little bit. For me your explanation is not clear, you explain all looking only the specular point but reflected signal depend on the Fresnel surface. This Fresnel surface is large for low elevation satellite and the corresponding height is an average of the top of the waves and the wave troughs.

               For high elevation satellite, this surface is smaller and can show the sea state with some times a height obtained at the wave top and sometime in the wave trough.

             May be add a figure that shows the linear shape of Ws and nonlinear shape of this wst function and you can explain why?

For 2.2 Experimental Equipment’s

M: You explain that you used a NEO M8T chipset with multi-constellation possibilities but you look only GPS signals why do you not used at least GLONASS, BEIDOU constellations? You reduce the number of satellite for a given time using only GPS, why this choice?  Explain              

M: Antennas, unfortunately, are not perfect and a RHCP antenna can acquire also LHCP signals and reversely. Do you have integrate in the SNR analyses the antenna gain pattern (2D or better in 3D) for RHCP component and LHCP? The SNR amplitude is also affected by the gain depending on the satellite elevation you do not discuss this important point why? May be you do not have it (it is possible it is not easy to obtain this strategic information) but in that case you can discuss its impact in terms of height retrieval. 

For 2.3 Flight experiments

m:  line 91 : replace "as an on situ  reference station" by "as an in situ base station"

m: line 92: add after height "using DGPS technic" and for your base station can you give the height error!

m: line 94: after which add "represents a base line of ~1km"

m: line 95 replace altitude by height

m: line 96 replace "to this on-site station" by "to our base station" 

m: figure 3a add a visible north Arrow and a scale bar because coordinates are to small!, figure 3b in the caption add the antenna height of your base station (I suppose it is your base station on the picture!).

For 3 results

m: you do not totally describe the two time series: you  give the mean values in the table 1, but do you have a change in the RMS for constant altitude flight and descending/ascending parts?

For GNSS-R during ascending or descending parts the RMS seems lower than constant parts why? Is it the same for H?

M: you have for the two flights almost 4 GPS satellites, but if you used all the constellations do you obtained better results?

For 4 discussion

M: You must improve this section. Put the description of the weight function in the theoretical part. A more interesting discussion can be first a comparison of the various weight function, a second part could be a comparison with other GNSS-R altimetry methods and in the last step with radar altimetry satellite. And you can finish by demonstrating the interest of this method for coastal areas for example. 

And a short real conclusion must be added.

Author Response

Thank you very much for your constructive comments.

They were very useful to revise our manuscript.

Basically, we follow most of your comments.

Please see the attached documents for details of our revisions.

Reviewer 3 Report

THis paper used two seperate receivers to get the direct and reflected signals, but the synchronized is quite important specially for the 1 cm level accuracy.

The samples points are too much less.

The height accuracy is not analyzed in detail.

Author Response

Thank you very much for your comments.

We do agree that the synchronized receivers is quite important.

However, we could not find light-weight GNSS receivers whose cost is low enough since the UAV might submerge into the lake.

Therefore, we have changed the title to declare we dare to use two classical recivers, and explain how we can obtain estimations even by two independent receivers.

Also in Section 4, we have added the estimated Delta T, indicating how two unsynchronized receivers suffer Delta T if they are not treated as the present method.

We also agree that the samples points are limited. Therefore we include discussions in Section 4 that possible modifications that may required when the present method is applied to other site, such as open oceans with larger SWHs.

Round  2

Reviewer 2 Report

First, congratulation for this work improvement, you have answered to a main part of my questions.

I have just a few remarks:

line 91-105 ok for me now.

your answer: " LHCP/RHCP ratio exceeds 15 dB" is ok but I think it is for high elevation satellite and in the case, as you said, "the atmospheric contamination is almost the same for the direct and the reflected path".

But take care,  at low elevation LHCP/RHCP ratio, I think, is maller due to the fact that the ray do not arrived on the maximum gain of the antenna, other point the direct and the reflected path are quite diffrent. Add in your text some limitation for very low elevation.

line 116-119: It is more clear now and I unbderstand why you used only GPS signal.

line 136 suppress "aproximately 1km"  and put ~1km.

line 272 suppress the setence : "this recursive estimation method will be reported in another paper"

It could be better if you give an idea how you obtain it?

line 280 may be you can add "but it was observed by static GNSS-R analysis (Roussel et al., 2016)" 

N. Roussel, F. Frappart, G. Ramillien,J. Darrozes, C. Desjardins, P. Gegout, F. Pérosanz, and R. Biancale (2015), Sea level monitoring and sea state estimate using a single geodetic receiver , RSE. 

Improve your summary including some perspective like the analysis of surges, waves in coastal areas, floods.

Author Response

Thank you very much for your valuable comments. 

We have revised our manuscript based on all of your comments.

-----For line 91-105, we have modified the explanation on LHCP/RHCP ratio as 

"Both antennas have reasonable LHCP versus RHCP gain ratios by larger than 15 dB down to 10-deg elevation angle, so we limit the elevation angle of GNSS satellites to be larger than 10-deg." 

-----For line 136, the text has been modified as suggested.

-----For line 272, the text has been modified as suggested.

  In addition, we have added some descriptions what is remained to be studied in future.

-----For line 280, the text has been modified as suggested.

-----For the summary, we have extended to include some perspectives.

-----Moreover, we have added additonal discussions on the tendency of the overestimated h at higher UAV altitudes. 

The path length tends to be larger than the assumed geometric one if the actual reflection point is displaced by the effect of tilting slopes, since the minimum path length is realized at the geometric reflection point. By comparing the weight functions, we have revealed that the effect of tilting slope is enhanced at higher altitudes, and thus the obtained h tends to be overestimated at higher UAV altitudes. 

I hope these revisions satisfy your requests.

Reviewer 3 Report

1. On the basis of traditional GNSS-R altimetry measurement, this paper proposes a weight function. The weight value is a function of the elevation angle of the GNSS satellite, and uses different weight values to solve it by least squares. In the result analysis part of the paper, the calculation accuracy of different weight values is analyzed by actual data. But this part is the focus of the model, please give the relevant theoretical analysis in Section II.

2. In the experimental part, does the attitude change of UAV affect the result? How to calibrate the flight altitude of UAV? Please explain on it.

Author Response

Thank you for your comments, which were really helpful to realize us additional facts.

Basically, we have modified our manuscript based on all of your comments.

1. We have modified our discussion section to explain the discrepancy of the statistics from view points of the theretical basis of the weight functions. The displacement of the reflection point due to the presence of tilting slopes (for w^st) is significant at the higher UAV altitudes. Meanwhile, the weitht w^s tends to eliminate GPS satellites with lower-elevation angles, and thus the statistics becomes less sensitive to the actual GPS elevation angles in two flights.

2. The effect of the attitude change of the UAV was explained in a paragraph below Fig.4 (L164-168). 

The flight altitude of the UAV (Ha) was determined by 5 Hz PPK positioning (Section 2.3), which shows smooth changes without inconsistent gaps.

This remided us that the negatively-larger tendency of the mean difference at the higher UAV altitudes in Table 1 is caused by the overestimated h, not by error of Ha. The displacement of the actual reflection point by tilting slopes does overestimate h, and which becomes significant at higher UAV altitudes, as described above. We have added this explanation in Discussion Section.

We hope these revisions satisfies your comments.